# Identification and Characterization of Viral and Bacterial Pathogens in Free-Living Bats of Kopaonik National Park, Serbia

**DOI:** 10.3390/vetsci12050401

**Published:** 2025-04-24

**Authors:** Dejan Vidanović, Nikola Vasković, Marko Dmitrić, Bojana Tešović, Mihailo Debeljak, Milovan Stojanović, Ivana Budinski

**Affiliations:** 1Veterinary Specialized Institute Kraljevo, 36000 Kraljevo, Serbia; 2Department of Genetic Research, Institute for Biological Research “Siniša Stanković”—National Institute of the Republic of Serbia, University of Belgrade, 11108 Belgrade, Serbia

**Keywords:** chiroptera, coronavirus, *Mycoplasma*, *Rickettsia*, viruses, bacteria, zoonoses

## Abstract

Globalization, climate change, and increasing human-driven environmental changes have elevated the risk of zoonotic disease emergence—diseases that can be transmitted from animals to humans. Bats, while playing vital ecological roles, are recognized as natural reservoirs of numerous pathogens, including over 200 viruses, such as lyssaviruses, filoviruses, coronaviruses, and henipaviruses, some of which are linked to serious diseases like rabies, Ebola, and SARS-CoV. Although viral diversity in bats is well-documented, bacterial pathogens remain underexplored, despite growing evidence of their presence. This pilot study investigated 40 individuals from 12 bat species in Serbia’s Kopaonik National Park using microbiological and molecular methods. While no high-risk pathogens, such as SARS-CoV-2, lyssaviruses, or *Salmonella* spp., were found, alphacoronavirus genomes were confirmed in four bats. Additionally, genomes of *Mycoplasma* spp. were present in 45% and *Rickettsia* spp. in 12.5% of individuals, although species-level identification of these pathogens was not possible. These findings highlight the need for continued surveillance of bat-associated microorganisms, particularly in areas where human–wildlife interactions are increasing. Understanding the microbial diversity in bat populations is essential for anticipating potential zoonotic threats and informing public health strategies.

## 1. Introduction

Globalization and environmental, climate, and human-induced changes provide ample opportunities for the spread and emergence of zoonoses—diseases that can be transmitted between animals and humans. Today, it is known that about 75% of all infectious diseases in humans are zoonoses, resulting from the transmission of pathogens from domestic or wild animals to humans [1].

Bats (order Chiroptera) are the second most numerous group of mammals in the world, right after rodents. More than 1480 species have been described so far [2], and bats account for more than 20% of all mammalian species. They play an important role in ecosystems worldwide, as suppressors of pest insect populations, seed dispersers, pollinators, etc. However, alongside these ecological benefits, bats have also been increasingly recognized for their role in the transmission of infectious diseases. Growing evidence suggests that bats are significantly underestimated as natural reservoirs of a wide range of viruses, including many of public health significance [3]. Nevertheless, they have a similar proportion of viruses per host species as other orders of mammals [4,5].

To date, more than 200 viruses have been identified in bats, including species from both suborders of Chiroptera: Yinpterochiroptera (including the superfamily Rhinolophoidea and family Pteropodidae) and Yangochiroptera (which includes all other bat families). These viruses belong to 27 different viral families, indicating an extraordinary diversity in bats of virus species [6], some of which are highly pathogenic to humans. Adding to the need for precaution is that bats serve as hosts for the recombination, transmission, and spread of viral pathogens.

Perhaps the best-known example of viruses associated with bats is the rabies virus (family *Rhabdoviridae*), with a link between bats and human infections recognized for over a century—since the first identification of rabies in asymptomatic vampire bats [7]. Other important groups of bat-associated viruses include henipaviruses (Nipah virus and Hendra virus), filoviruses (Marburg virus, Ebola virus, and Mengla virus), bunyaviruses (e.g., Ahun virus), and coronaviruses, such as SARS-CoV and MERS-CoV. SARS-CoV emerged in 2002–2003 in Guangdong Province, China, and bats were later identified as the natural reservoir of this novel pathogen. A decade later, in 2012, MERS-CoV emerged in the Middle East. Although dromedary camels were the direct source of human infections, viruses closely related to MERS-CoV have also been detected in bats, suggesting a possible ancestral origin in bat populations [8].

While global reviews have highlighted the roles of bats as carriers and potential reservoir hosts of human-pathogenic viruses, much less is known about the public health relevance of viruses found specifically in European bat populations. European bats have been increasingly recognized as hosts of a wide diversity of viruses, some of which have zoonotic potential. A comprehensive review by Kohl et al. [9] identified viruses from 21 different viral families in European bats, with *Adenoviridae*, *Rhabdoviridae*, and *Coronaviridae* being the most frequently detected and most intensively studied. Among these, lyssaviruses—particularly European Bat *Lyssavirus* 1 (EBLV-1) and European Bat *Lyssavirus* 2 (EBLV-2)—remain the only bat viruses in Europe with clearly established zoonotic potential, capable of causing fatal rabies-like disease in humans, although spillover events are rare. Molecular surveillance has also revealed the presence of various coronaviruses (mainly alpha- and betacoronaviruses) and novel paramyxoviruses, but none have been linked to human disease so far [9].

In contrast to viruses, bacterial pathogens in bats have been understudied despite their ubiquity and diversity. However, in recent years, this trend has been changing, with an increasing number of studies investigating bacteria (from vector-borne to enteric pathogens), protozoa, parasites, and fungal agents present in bats. The most prevalent pathogens are from the genera *Leptospira*, *Mycoplasma*, *Brucella*, *Coxiella*, *Francisella*, and *Rickettsia* [10]. The dynamics of infection in bats is driven by a complex interaction of environmental, immunological, behavioural, and anthropogenic factors [11]. In the future, it will be crucial to conduct interdisciplinary studies to better understand the circumstances that contribute to the emergence of disease in bat populations and to reduce threats to humans and animals.

To date, 31 species of bats have been recorded in Serbia, indicating the great diversity of this group of mammals in the country [12]. While in some parts of Serbia, bat surveys have been systematically conducted over many years, data on bat fauna are sparse in other areas (including many parts of Central Serbia), with little or no information available on the pathogens associated with bats in Serbia. Remarkably, there is a lack of published studies on bat pathogen investigations within the country.

Kopaonik National Park covers the highest and most well-preserved parts of the Kopaonik mountain, which rises in the central part of southern Serbia. The park covers an area of 11,810 hectares and begins at an altitude of about 800 m and ends with the highest peak 2017 m above sea level. Due to its natural beauty and high level of biodiversity, Kopaonik National Park is a popular tourist attraction, so it could be a hotspot of interaction between bats and humans.

The aim of this pilot study was to detect and classify pathogenic microorganisms in bat species inhabiting the Kopaonik National Park area in order to improve knowledge about the pathogens’ distribution and to create control and surveillance measures for their potential transmission to humans.

## 2. Materials and Methods

### 2.1. Study Area and Sample Collection

During the summer and autumn of 2024 (8–9 July 2024, 21–22 August 2024, and 10–11 October 2024), three field studies were conducted in Kopaonik National Park, Serbia. Bats were trapped at several sites at the locations Metođe (geo: 43.2694, 20.8247, altitude 1540 m) and Zaplanina (geo: 43.2651, 20.8598, altitude 1104 m). Before trapping, acoustic monitoring across various sites within the park, covering different habitats and altitudes, was conducted. Based on the areas with the highest bat activity and species diversity revealed through acoustic surveys, specific trapping sites were selected. Trapping was carried out at the entrances of mine shafts (where a higher number of bat passes were recorded) as well as along forest trails where bat activity was also detected. For trapping, standard vertical nets (mist-nets, Ecotone, Poland) 3–12 m in length were set up near entrances to abandoned mine shafts and across forest trails. Nets were opened at sunset and taken down 4 h later. Trapped bats were identified to the species level following Dietz and Kiefer [13], sexed, and their age and reproductive status were assessed. Four swabs (two oral and two anal) were collected from all individuals for bacteriological and virological analysis. Bats were released immediately after processing and sample collection at the trapping sites. Trapping and sample collection were carried out under permit no. 000493134 2023 14850 004 003 501 086 issued by the Ministry for Environmental Protection of the Republic of Serbia.

### 2.2. Bat Pathogen Analyses

Analyses of pathogens were carried out at the Veterinary Specialized Institute Kraljevo. The examined pathogens were as follows:

(1) Viruses: lyssaviruses (family *Rhabdoviridae*); henipaviruses (Nipah virus and Hendra virus); Lloviu virus (family *Filoviridae*); alpha-, beta-, gamma-, and deltacoronaviruses; and SARS-CoV-2 (family *Coronaviridae*).

(2) Bacteria: Salmonella, Leptospira, Mycoplasma, Coxiella, Brucella, and Rickettsia.

Swabs were kept on dry ice until they were brought to the laboratory, where they were stored at −80 °C until the beginning of the examination. For each bat, one oral and one anal swab were pooled and used for *Salmonella* isolation, while the second oral and anal swabs were pooled and then used for DNA/RNA extraction and molecular investigation.

*Salmonella* was isolated following the accredited method EN ISO 6579 (ISO 6579-1:2017, “Microbiology of the food chain—Horizontal method for the detection, enumeration, and serotyping of *Salmonella*”) [14]. Briefly, swabs were first cultured in pre-enrichment buffered peptone water (BPW) for 18 ± 3 h at 37 °C. Incubated pre-enrichment medium (0.1 mL) was transferred to modified semi-solid Rappaport–Vassiliadis (MSRV) agar and incubated at 40.5–42.5 °C. After 24 h incubation of selective enrichment, bacterial growth from the edge of the growth zone was streaked onto plates, one containing xylose–lysine–deoxycholate (XLD) agar and one brilliant green agar (BGA).

For genomic DNA and RNA extraction, oral and rectal swabs from each bat were placed together in 1.5 mL tubes with the addition of 800 µL of phosphate-buffered saline. After vortexing and brief centrifugation, 200 µL of supernatant was transferred to a fresh tube and extracted using a Bioextract Superball extraction kit (Biosellal, Dardilly, France) following the manufacturer’s instructions. A Luna universal probe One Step RT-PCR kit (NEB, Ipswich, MA, USA) was used for real-time and RT-PCR reactions, Luna universal RT-PCR mastermix (NEB, USA) was used for SYBR RT-PCR reactions, and Luna universal probe qPCR master mix was used for real-time PCR reactions. Real-time PCR reactions were performed in an AriaMx thermal cycler (Agilent, Penang, Malaysia), and RT-PCR reactions were performed in a 2720 thermal cycler (Applied Biosystems, Singapore).

The presence of lyssaviruses was determined by using the Pan-lyssavirus Real Time RT-PCR assay for Rabies Diagnosis [15]. Filoviruses were determined using two assays, RT-PCR described by He et al. [16] and SYBR RT-PCR described by Coertse et al. [17]. Henipaviruses and Nipah and Hendra viruses were determined using RT-PCR assays described in [18]. Coronaviruses were determined using two assays, i.e., a nested RT-PCR assay [19] and a RT-PCR assay [20], and SARS-CoV-2 virus was determined using the real-time RT-PCR assay developed by Corman et al. [21]. Mycoplasmas were determined using the PCR assay described by Cultrera et al. [22]; Chlamydiaceae were determined using the real-time PCR assay described by Ehricht et al. [23]; *Coxiella burnetii* was determined using the real-time PCR assay described by Klee et al. [24]; and Rickettsiae were determined using the real-time PCR assay described by Kato et al. [25]. Pathogenic *Leptospira* were determined using the Real Time PCR assay described by Stoddard et al. [26]. Primers and probes used in this study are listed in Table 1.

PCR products of coronaviruses and mycoplasmas were purified and sequenced with a Sanger SeqStudio sequencer (ThermoFisher Scientific, Waltham, MA, USA) externally. The obtained sequences were processed in Chromas lite 2.1 software and compared with other coronavirus sequences in National Centre for Biotechnology Information (NCBI, Betesda, ML, USA) database using free online Basic Local Alignment Search Tool (BLAST) software. Mega 12 software was used for phylogenetic analysis.

## 3. Results

A total of 40 bats from 12 different species were captured and sampled: *Rhinolophus ferrumequinum*, *Rhinolophus hipposideros*, *Barbastella barbastellus*, *Myotis alcathoe*, *Myotis blythii*, *Myotis brandtii*, *Myotis cf. mystacinus*, *Myotis daubentonii*, *Myotis emarginatus*, *Myotis myotis*, *Myotis nattereri*, and *Plecotus auritus*.

*Salmonella* was not isolated by the culture method from any of the bats. The genomes of lyssaviruses, filoviruses, henipaviruses (Hendra and Nipah), and SARS-CoV-2 viruses were not detected using molecular methods. Coronavirus genomes were detected in 4 of 40 bats, 1 in the bat species *Myotis brandtii* (No. 4), 2 in *Myotis daubentonii* bats (No. 6 and No. 8), and 1 in a *Myotis* cf. *mystacinus* bat (No. 35) (Table 2). Coronaviruses in bats No. 4 and No. 6 were detected using the Holbrook nested RT-PCR assay [19], and coronaviruses in bats No. 8 and No. 35 were detected using Watanabe RT-PCR assay [20]. Sequencing of part of the coronavirus genome from bat No. 4 (271 nucleotides) classified it as a bat alphacoronavirus, with 98.56% similarity to a virus from Russia (Bat-CoV/M.brandtii/Russia/MOW15-27/1, OQ725983.1) detected in *M. brandtii*. The same was true for the genome from bat No. 6 (337 nucleotides), which also showed 98.56% similarity to the same virus from Russia. The genomes of the viruses from bats No. 4 and No. 6 differed by only one nucleotide (99.6% similarity). Bat coronavirus from bat No. 8 had 99.20% similarity with a bat coronavirus isolate from Denmark, BtCoV/13585-58/M.dau/DK/2014 (MN535732.1), detected in *M. daubentonii*, while bat coronavirus from bat No. 35 had 93.92% similarity with a bat coronavirus isolated from Russia, BtCoV/631/RUS/2020 (MZ322309.1), detected in *M. daubentonii*.

Phylogenetic analysis (Figure 1) of part of the RdRp gene showed that the alphacoronaviruses detected in bats No. 8 and No. 35 were classified in the subgenus *Pedacovirus*, while the viruses detected in bats No. 4 and No. 6 were classified in the subgenus *Myotacovirus*.

The evolutionary history was inferred using the Neighbor-Joining method. The optimal tree with the sum of branch length = 0.656 is shown. The percentages of replicate trees in which the associated taxa clustered together in the bootstrap test (1000 replicates) are shown next to the branches. The tree is drawn to scale, with branch lengths in the same units as those of the evolutionary distances used to infer the phylogenetic tree. The evolutionary distances were computed using the Maximum Composite Likelihood method and are in units of the number of base substitutions per site. The analytical procedure encompassed 48 coding nucleotide sequences using the first, second, third, and non-coding positions. The pairwise deletion option was applied to all ambiguous positions for each sequence pair, resulting in a final data set comprising 304 positions. Evolutionary analyses were conducted in MEGA12 [27]. Sequences of alphacoronaviruses from Serbia are labelled with red diamond.

*Mycoplasma* genomes were detected in 18 of 40 bats (45%), the species being *R. ferrumequinum, M. brandtii, M. daubentonii, M. blythii, M. myotis, M. emarginatus, M.* cf. *mystacinus,* and *P. auritus*. Nucleotide sequencing of PCR products could not reveal the species of *Mycoplasma* but only the genera. *Rickettsia* species genomes were detected in 5 of 40 bats (12.5%), with the species being *M. brandtii, M.* cf. *mystacinus, B. barbastellus*, and *P. auritus*. The genomes of other pathogenic bacteria, i.e., *Chlamydia* spp., *Coxiella burnetii*, pathogenic *Leptospira* spp., and *Francisella tularensis*, were not detected in the examined bats. An overview of the microorganism genomes confirmed in the analysed bats is given in Table 2.

## 4. Discussion


*Coronaviruses*


Coronaviruses (order Nidovirales, family *Coronaviridae*) include four genera: alphacoronavirus and betacoronavirus, which infect a wide range of mammals, and gammacoronavirus and deltacoronavirus, which primarily infect birds. Coronaviruses have been identified in 238 species of bats worldwide. Alphacoronaviruses and betacoronaviruses have been identified in bats from 14 of the 21 bat families, in at least 69 countries on six continents [28]. In Europe, coronaviruses have been recorded in the bat families Rhinolophidae, Vespertilionidae, and Miniopteridae, in nine genera (*Rhinolophus*, *Myotis*, *Pipistrellus*, *Plecotus*, *Tadarida*, *Nyctalus*, *Hypsugo*, *Eptesicus*, and *Miniopterus)* and, in total, in 26 species [29], including *Myotis brandtii* and *Myotis daubentonii*. Phylogenetic analysis of alphacoronaviruses detected in Kopaonik National Park classified them as belonging to two subgenera of alphacoronaviruses, *Myotacovirus*, detected in *M. brandtii* and *M. daubentonii*, and *Pedacovirus*, detected in *M. daubentonii* and *M.* cf. *mystacinus.* Our literature review did not reveal any previous studies confirming the presence of alphacoronaviruses in *M. mystacinus* bats. These results are in accordance with the results published previously by researchers from the Scientific Institute of Veterinary Medicine Novi Sad, Serbia, who determined the presence of alpha and betacoronaviruses in 35 out of 142 samples in the territory of Vojvodina province (Serbia) [30].

Alphacoronaviruses mainly infect bats, but they have the potential to transfer to humans or other animals and cause disease. Although these viruses have not yet caused significant outbreaks among humans, they are considered a potential organism for future zoonotic transmission [31], particularly in areas where there is close contact between humans and bats (e.g., caves) or habitats with a high abundance of bats. Recent research shows that all currently known human coronaviruses have an animal origin, with bat or rodent coronaviruses being the most probable ancestors. In most cases, it was suggested that other mammals served as intermediate hosts prior to final adaptation to humans [32].

Analysis of the genome sequence identity and geographic distribution of the homologous alphacoronaviruses examined in this pilot study in Kopaonik National Park provides valuable information into the broad geographic distribution and evolutionary dynamics of these viruses. Alphacoronaviruses from bats in Kopaonik National Park had high sequence identity (up to 99%) with bat alphacoronaviruses from Russia, Denmark, Finland, and Germany; this is in agreement with results obtained by other researchers [33,34]. This extensive geographic distribution highlights the ecological adaptability of alphacoronaviruses and their potential to spread across large and diverse regions, most likely via the short- and long-distance movements of bats [34]. Bats are capable of flight, enabling them to maintain and spread the pathogens they carry. These pathogens include vector-borne parasites and bacteria, which can also be transmitted through ectoparasites such as bat flies and ticks [35].

The investigation and monitoring of coronaviruses is crucial in order to detect in time possible mutations that could increase their transmissibility or pathogenicity from bats to humans.


*Lyssaviruses, Filoviruses*


Neither lyssavirus nor filovirus genomes were detected in the 40 bats from Kopaonik National Park. The results of this study are in accordance with earlier research on lyssaviruses in bats in Serbia, where these viruses were not detected. The only reported case of lyssavirus in bats in Serbia was in 1954 [36], but since then, there have been no reported cases. In 2006, a two-year study of bats did not reveal any positive case [37]. Elsewhere in Europe, lyssaviruses have been detected in several countries, including the United Kingdom, the Netherlands, Finland, Germany, Spain, Italy, Slovenia, Croatia, Hungary, Bulgaria, Ukraine, and Russia. As expected, filovirus genomes, such as Ebola virus and Marburg virus, were not found in any bat from Kopaonik National Park, as these viruses are associated with bats from the family Pteropodidae that is not present in Serbia. Lloviu virus genome was also not detected in the Kopaonik bats, although it was previously detected in *Miniopterus schreibersii* bats in other locations in Serbia, neighbouring Bosnia and Herzegovina, and Hungary [38]. *M. schreibersii* is the main host of this filovirus [39], but during the current research, no bats of this species were captured in Kopaonik National Park.

*Mycoplasma* spp.

The occurrence of mycoplasmas in bats has potential ecological and epidemiological significance for both human and domestic animal health. Mycoplasmas are distributed worldwide and are present in various species of bats, although a higher prevalence of mycoplasmas has been observed in vampire bats (Desmodontinae) compared to other species [40]. Some *Mycoplasma* species are zoonotic, meaning they can be transmitted from bats to humans or domestic animals. Given that mycoplasmas frequently colonize the respiratory and urogenital tracts, their role in causing disease in domestic animals, such as cattle, pigs, horses, or poultry, is possible. Mycoplasmas have already been recognized as causative agents of diseases such as swine enzootic pneumonia (*Mycoplasma hyopneumoniae*) or bovine infectious pneumonia (*Mycoplasma bovis*). Contact between bats and domestic animals, especially in rural areas, increases the risk of transmission. In some studies, the prevalence of *Mycoplasma* in bats ranged from 3.2% [41] up to 47% [42]. Although mycoplasmas generally cause mild or opportunistic infections in humans, certain species, such as *Mycoplasma pneumoniae*, the cause of atypical pneumonia, can cause more serious problems [35].

Among the 40 bats from Kopaonik National Park studied, 18 (45%) harboured mycoplasma genomes: *Rhinolophus ferrumequinum* (6×), *Plecotus auritus* (3×), *Myotis myotis* (3×), *Myotis emarginatus* (2×), *Myotis blythii* (1×), *Myotis brandtii* (1×), *Myotis* cf. *mystacinus* (1×), *Myotis daubentonii* (1×). Findings of mycoplasmas and their genomes in bats indicate their potential importance as reservoirs for these bacteria, requiring further research to understand the risks to humans and domestic animals. Monitoring and studying these microorganisms can contribute to better prevention of zoonotic infections and preservation of human and animal health.

*Rickettsia* spp.

Bacteria of the genus *Rickettsia* are intracellular pathogens transmitted by vectors, such as ticks, fleas, and lice, and are known to cause a variety of diseases in humans and animals, including rickettsioses (such as typhus and spotted fever). The presence of *Rickettsia* in bats is of potential importance to human and domestic animal health due to the possibility of zoonotic transmission, although this area is not yet sufficiently investigated. If bats serve as reservoirs for *Rickettsia* bacteria, there is a potential risk that the bacteria could be transmitted to humans by ectoparasites living on bats. Given that bats are common in urban and rural areas, contact with their habitats or ectoparasites may increase the possibility of outbreaks of zoonotic diseases. The presence of *Rickettsia* in bats could also pose a risk to domestic and wild animals, especially those that share a habitat with bats or are in contact with the same vectors. There are currently not many data on the prevalence and species of *Rickettsia* in bats, and even though these potentially zoonotic bacterial pathogens are frequently detected in bat-associated ectoparasites, their role in pathogen transmission remains poorly understood [40]. *Rickettsia* genomes were detected in five of 40 bats (12.5%) from Kopaonik National Park: *Plecotus auritus* (1×), *Myotis brandtii* (2×), *Myotis* cf. *mystacinus* (1×), and *Barbastella barbastellus* (1×). Our findings are in agreement with the findings of researchers from Slovenia who reported the presence of rickettsiae in 17% of bats [43] but, unlike their study, where the highest prevalence of *Rickettsia* was found in the genus *Rhinolophus*, our study did not detect *Rickettsia* in this genus but rather in the genus *Myotis*. Since the aforementioned study employed NGS metagenomic analysis of faecal samples, the authors concluded that caution should be exercised when interpreting the results, as identification was achieved at the *Rickettsia* genus level rather than the species level. They recommended species-specific PCR for further identification of *Rickettsia* [41]. A similar study was conducted in Romania [44], where tissue samples from dead bats were used. Species-specific PCR was performed and 17 sequenced samples had 99.7–100% genetic identity to *Rickettsia monacensis* strains from *Ixodes ricinus* ticks in Romania, Italy, and Serbia. Of the 17 sequenced samples, 5 were from *Pipistrellus pipistrellus* and 12 from *Nyctalus noctula*. That study [44] reported the first detection of *R. monacensis* in European bat tissues and the first global record in these two bat species. While Spotted Fever Group (SFG) rickettsiae have previously been found in ectoparasites associated with bats, their presence in bat tissues suggests that bats may play a role in the transmission or maintenance of *Rickettsia* [44]. The presence of *Rickettsia* genomes in bats in Kopaonik National Park highlights the need for further research. The first step would be to sequence the genomes obtained in order to identify the exact species, followed by attempts to determine the role of bats as a reservoir and the possibility of zoonotic transmission of *Rickettsia*. A better understanding of transmission dynamics, as well as vector identification, is essential for risk assessment and the development of preventive measures. This includes monitoring bat habitats, vector control, and education about potential risks in communities that come into close contact with bats.


*Other potentially pathogenic bacteria in bats*


*Salmonella* was not isolated from the 40 bats from Kopaonik National Park. However, several members of the Enterobacteriaceae family, including *Salmonella*, have previously been isolated from bats. Studies have shown that *Salmonella* serotypes isolated from bats have similar characteristics to those found in domestic animals and humans, suggesting that bats may be locally important in the epidemiology of salmonellosis in humans and domestic animals [45]. Some of the serotypes isolated from bats, i.e., *S. Typhimurium*, *S. Enteritidis*, and *S. Virchow*, are common causes of disease in humans and animals [46].

*Coxiella*, *Leptospira*, and *Chlamydia* genomes were not detected in the 40 Kopaonik bats. *Coxiella burnetti* has previously been reported in bats, but the role of these animals in the spread of *Coxiella* has not yet been fully investigated [47]. Bats have recently been identified worldwide as an important reservoir of various *Leptospira* species (*L. interrogans*, *L. borgpetersenii*, *L. kirschneri*, *L. fainei*, and *L. noguchii*). Currently, more than 107 species of bats have been reported to harbour these bacteria [48], but as prevalences range from 0% to 85.7% [40], the absence of signals from *Leptospira* in the Kopaonik bats is unsurprising. Chlamydiae have not been studied in bats in Serbia to date, so this report on the absence of their signals in the Kopaonik bats is new to the scientific record.

This study presents the first results of microbiological research on bats in Kopaonik National Park, contributing to a broader understanding of the presence of pathogenic microorganisms in the region. However, some limitations of this study must be considered. The sample size was relatively small, primarily due to the high altitude of the National Park. Previous knowledge about bat distribution within Kopaonik National Park was scarce prior to this research, and no data on known bat roosts were available. To survey bat presence, potential roost sites (such as abandoned buildings and mine shafts) were searched, since there are no natural caves in the area. Given the high altitude of the study area, nursery colonies were not expected to be found, even though sampling was during the summer and autumn (females of some bat species form large nursery colonies during the reproductive period). This assumption was confirmed, as only adult individuals were trapped during the summer and autumn of 2024. It is likely that bats use the mine shafts as transitional and/or swarming roosts. There are some bat records from Serbia at the altitude of Kopaonik, but very few bat roosts are known at this elevation [12]. Therefore, since the presence of numerous colonies was not anticipated, we did not expect to capture a large number of individuals. However, despite the relatively low number of trapped bats (*n* = 40), species diversity was greater than expected (*n* = 12). This large species diversity, revealed for the first time in our pilot study, should stimulate further much-needed research into the bat populations in Kopaonik National Park.

Live animals were captured and not euthanized in our pilot study. No carcasses of dead bats were found during the sampling from which internal organs could have been collected, potentially containing pathogens that are not present in oral or rectal swabs. PCR-based screening, although highly sensitive, allows only known or closely related pathogens to be detected, potentially failing to capture more divergent or novel viruses.

This pilot study focused on targeted pathogenic microorganisms previously documented in the literature as animal or zoonotic pathogens. However, confirming and expanding these findings requires a larger number of samplings throughout the year to obtain a larger sample size, including ectoparasites such as bat flies and ticks. Future research should also include next-generation sequencing (NGS) metagenomic analysis to comprehensively characterize the microbial communities present in bats. Metagenomic approaches would enable unbiased detection of both known and novel pathogens.

## 5. Conclusions

Bats are of great importance in the preservation of the entire ecosystem, but at the same time, they can be reservoirs of various pathogens, including viruses and bacteria with zoonotic potential, which pose a potential threat to human and animal health. Bats from Kopaonik National Park harboured the genomes of some potentially zoonotic microorganisms, such as alphacoronaviruses, mycoplasmas, and rickettsiae, while genomes of other, more dangerous pathogens, such as coronaviruses related to SARS-CoV-2, lyssaviruses, filoviruses, and paramyxoviruses, were not found in the animals.

The results of the virological tests were expected, because so far, lyssaviruses and paramyxoviruses have not been detected in Serbia, primarily due to the absence of bat species that are their carriers. The finding of alphacoronaviruses were also expected, given their well-documented presence in Serbian and European bat populations.

Although this pilot study was conducted on a small number of bats from a relatively small geographical area, it is the first systematic pathogen surveillance in bats conducted in Kopaonik National Park and provides a good starting point for further research. The results highlight the need for continuous monitoring and longitudinal study of bats as reservoirs of pathogens. Implementing regular monitoring programs would significantly improve the country’s capacity for early detection of pathogens, which is crucial for improving the prevention of animal diseases and zoonoses and for supporting One Health initiatives aimed at protecting human and animal health.

## Figures and Tables

**Figure 1 vetsci-12-00401-f001:**
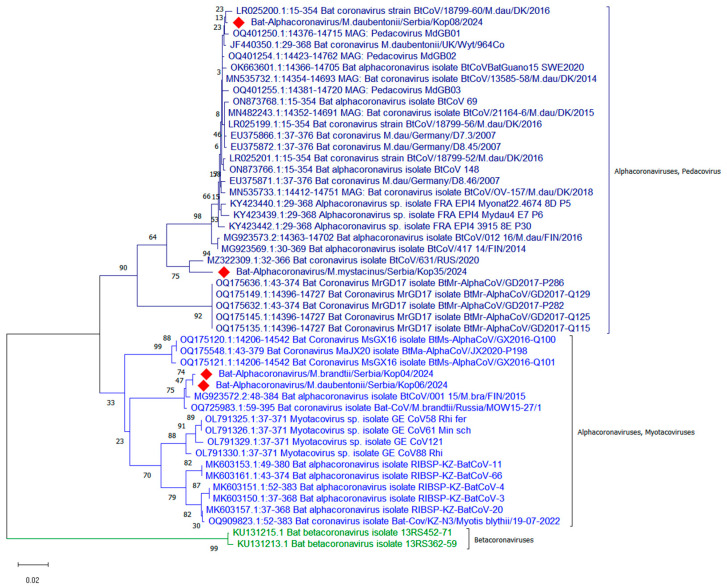
Evolutionary relationships of 48 taxa based on partial RdRp gene.

**Table 1 vetsci-12-00401-t001:** Primers and probes used in this study.

Assay	Primer/Probename	Primer/Probe Sequences
*Lyssavirus* [15]	JW12	ATG TAA CAC CYC TAC AAT G
	N165	GCA GGG TAY TTR TAC TCA TA
Filoviruses [16]	Filo-F	TGATATATGATCATCTTCCAGG
	Filo-in-F	GCATTTCACCAATTAACACAGG
	Filo-R	TTTATATGAATCAGTGGAGGTG
	FV-F1	GCMTTYCCIAGYAAYATGATGG
	FV-R1	GTDATRCAYTGRTTRTCHCCCAT
	FV-F2	TDCAYCARGCITCDTGGCAYC
	FV-R2	GIGCACADGADATRCWIGTCC
Filoviruses [17]	Filo-SYBR-F	GRGARTAYGCICCITTYGC
	Filo-SYBR-R	AGYTGYTGRTAYTGYTCICC
Hendra [18]	HeV M 5755F	CTTCGACAAAGACGGAACCAA
	HeV M 5823R	CCAGCTCGTCGGACAAAATT
Nipah [18]	NiV_N_1198F	TCAGCAGGAAGGCAAGAGAGTAA
	NiV_N_1297R	CCCCTTCATCGATATCTTGATCA
Pan-corona [19]	Pan_CoV_F1	GGTTGGGAYTAYCCHAARTGYGA
	Pan_CoV_R1	CCRTCATCAGAHARWATCAT
	Pan_CoV_R2	CCRTCATCACTHARWATCAT
	Pan_CoV_F2	GAYTAYCCHAARTGTGAYAGA
	Pan_CoV_F3	GAYTAYCCHAARTGTGAYMGH
Coronavirus [20]	Watanabe conventional_F	GGTTGGGACTATCCTAAGTGTGA
	Watanabe conventional_R	CCATCATCAGATAGAATCATCATA
SARS-CoV2 [21]	SARS2-IP4-14059F	GGT AAC TGG TAT GAT TTC G
	SARS2-IP4-14146R	CTG GTC AAG GTT AAT ATA GG
	SARS2-IP4-14084FAM	FAM-TCA TAC AAA CCA CGC CAG G-BHQ1
*Mycoplasma* [22]	GPO-1	ACTCCTACGGGAGGCAGCAGTA
	MGSO	TGCACCATCTGTCACTCTGTTAACCTC
Chlamydiae [23]	Ch23S-F	CTGAAACCAGTAGCTTATAAGCGGT
	Ch23S-R	ACCTCGCCGTTTAACTTAACTCC
	Ch23S-Pro	FAM-CTCATCATGCAAAAGGCACGCCG-BHQ1
*Coxiella burnetii* [24]	Cox-F	GTCTTAAGGTGGGCTGCGTG
	Cox-R	CCCCGAATCTCATTGATCAGC
	Cox-TM	FAM-AGCGAACCATTGGTATCGGACGTT-TAMRA-TATGG
*Rickettsia* [25]	PanR8_F	AGCTTGCTTTTGGATCATTTGG
	PanR8_R	TTCCTTGCCTTTTCATACATCTAGT
	PanR8_P	FAM-CCTGCTTCTATTTGTCTTGCAGTAACACGCCA-BHQ1
Pathogenic *Leptospira* spp. [26]	LipL32-45F	AAGCATTACCGCTTGTGGTG
	LipL32-286R	GAACTCCCATTTCAGCGATT
	LipL32-189P	FAM-AAAGCCAGGACAAGCGCCG-BHQ1

**Table 2 vetsci-12-00401-t002:** Pathogen genome presence in bats from Kopaonik National Park, Serbia.

Bat No.	Bat Species	Coronaviruses	*Mycoplasma* sp.	*Rickettsia* sp.
1	*Rhinolophus ferrumequinum*	−	+	−
2	*Rhinolophus ferrumequinum*	−	+	−
3	*Rhinolophus hipposideros*	−	−	−
4	*Myotis brandtii*	+	+	+
5	*Myotis brandtii*	−	−	−
6	*Myotis daubentonii*	+	+	−
7	*Myotis* cf. *mystacinus*	−	−	+
8	*Myotis daubentonii*	+	−	−
9	*Barbastella barbastellus*	−	−	+
10	*Myotis blythii*	−	+	−
11	*Myotis myotis*	−	+	−
12	*Rhinolophus ferrumequinum*	−	+	−
13	*Myotis daubentonii*	−	−	−
14	*Myotis nattereri*	−	−	−
15	*Myotis alcathoe*	−	−	−
16	*Myotis* cf. *mystacinus*	−	−	−
17	*Myotis brandtii*	−	−	+
18	*Myotis daubentonii*	−	−	−
19	*Rhinolophus ferrumequinum*	−	+	−
20	*Myotis myotis*	−	+	−
21	*Plecotus auritus*	−	−	+
22	*Plecotus auritus*	−	−	−
23	*Plecotus auritus*	−	−	−
24	*Myotis emarginatus*	−	+	−
25	*Myotis emarginatus*	−	+	−
26	*Myotis daubentonii*	−	−	−
27	*Myotis* cf. *mystacinus*	−	+	−
28	*Myotis daubentonii*	−	−	−
29	*Plecotus auritus*	−	+	−
30	*Myotis myotis*	−	−	−
31	*Rhinolophus ferrumequinum*	−	+	−
32	*Plecotus auritus*	−	+	−
33	*Plecotus auritus*	−	+	−
34	*Rhinolophus ferrumequinum*	−	+	−
35	*Myotis* cf. *mystacinus*	+	−	−
36	*Myotis myotis*	−	+	−
37	*Plecotus auritus*	−	−	−
38	*Barbastella barbastellus*	−	−	−
39	*Myotis daubentonii*	−	−	−
40	*Rhinolophus hipposideros*	−	−	−

## Data Availability

All data are contained within this paper.

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
