# Peer review of "Identification and Characterization of Viral and Bacterial Pathogens in Free-Living Bats of Kopaonik National Park, Serbia"

_vetsci, 2025, doi:10.3390/vetsci12050401_

Round 1
Reviewer 1 Report
Comments and Suggestions for Authors
Dear authors,
thanks for this interesting paper on an important topic.
Here are some recommendations to improve your paper:
- All number between 0 and 9 need to be spelled out. Please check in the whole document.
- All scientific name (species, genus...) needs to be in italics. Please change in the whole document
- Abstract and Introduction: < !--StartFragment -->Zoonotic disease refer to diseases that transmit from animal to human and vice versa therefore I would use “can be transmitted between animals and humans” and not restricting it to only animal to human.< !--EndFragment -->
- Line 130: changed "polled" by "pooled"
Thanks for making the necessary changes
Comments on the Quality of English Language- Review the whole paper and decide to write either in English UK or English American but not both.
Author Response
Dear reviewer,
We appreciate the time and effort that you dedicated to providing feedback on our manuscript and are grateful for the insightful comments on and valuable improvements to our paper.
We accepted all your suggestions and suggestions from other reviewers, we have proofread the entire paper by professional, and we think that the manuscript is now improved.
- All number between 0 and 9 need to be spelled out. Please check in the whole document.
We agree, all numbers are replaced.
- All scientific name (species, genus...) needs to be in italics. Please change in the whole document
We agree, all names are shanged.
- Abstract and Introduction: < !--StartFragment -->Zoonotic disease refer to diseases that transmit from animal to human and vice versa therefore I would use “can be transmitted between animals and humans” and not restricting it to only animal to human.< !--EndFragment -->
We agree, it is corrected.
- Line 130: changed "polled" by "pooled"
It is changed
Comments on the Quality of English Language
- Review the whole paper and decide to write either in English UK or English American but not both.
We agree, the manuscript has been proofread by professional.
Best regards,
Authors
Reviewer 2 Report
Comments and Suggestions for Authors
The manuscript entitled "Identification and Characterization of Viral and Bacterial Pathogens in Free-Living Bats of Kopaonik National Park, Serbia”, investigated the presence of potentially zoonotic microorganisms in bat species from the Kopaonik National Park, Serbia. There are major concerns which should be addressed.
- No need to write the first letter of keywords as capital letter except for the name of the bacteria.
- Line 53; please add comma before “etc” and add a dot after “etc”.
- Line 69; please add a dot after MERS-CoV.
- It would be better to add a table of the primers sequences used at the study.
- Did you sequence your positive PCR products?, you have to provide a section of sequencing and data analysis at your materials and methods section
- Lines 181-191; this should be relocated to be at the results section.
- I think the writing of the paper isn’t good enough with many English grammar mistakes.
- Discussion section writing is very poor and not well arranged, you have to rewrite again in a more professional way.
- The paper has only 1 table and no figures at all. I think you have to add more tables and add some figures either for the bats from which you collected your samples or any other figures related to your work.
- Why didn’t you provide photogenetic trees for the positive sequences you got?
- You have to discuss your results in a more professional way.
I think the writing of the paper isn’t good enough with many English grammar mistakes.
Discussion section writing is very poor and not well arranged, you have to rewrite again in a more professional way.
Author Response
Dear reviewer,
We appreciate the time and effort that you dedicated to providing feedback on our manuscript and are grateful for the insightful comments on and valuable improvements to our paper.
We accepted all your suggestions and suggestions from other reviewers, we have proofread the entire paper by professional, and we think that the manuscript is now improved.
- No need to write the first letter of keywords as capital letter except for the name of the bacteria.
It is corrected.
- Line 53; please add comma before “etc” and add a dot after “etc”.
It is corrected.
- Line 69; please add a dot after MERS-CoV.
It is corrected.
- It would be better to add a table of the primers sequences used at the study.
Table of the used primers is added.
- Did you sequence your positive PCR products?, you have to provide a section of sequencing and data analysis at your materials and methods section
Yes, we have sequenced coronaviruses and we added a paragraph in Material and methods section and Results section.
- Lines 181-191; this should be relocated to be at the results section.
We have relocated that part.
- I think the writing of the paper isn’t good enough with many English grammar mistakes.
We have engaged a professional proofreader and translator
- Discussion section writing is very poor and not well arranged, you have to rewrite again in a more professional way.
We have tried and rewrited the whole section
- The paper has only 1 table and no figures at all. I think you have to add more tables and add some figures either for the bats from which you collected your samples or any other figures related to your work.
We have added one table and one figure more.
- Why didn’t you provide photogenetic trees for the positive sequences you got?
We have provided a phylogenetic tree for coronaviruses.
- You have to discuss your results in a more professional way.
We have tried and we have rewrited that section.
Comments on the Quality of English Language
I think the writing of the paper isn’t good enough with many English grammar mistakes.
Discussion section writing is very poor and not well arranged, you have to rewrite again in a more professional way.
We agree, the manuscript has been proofread by professional and we beleave that now it is much better.
Best regards,
Authors
Reviewer 3 Report
Comments and Suggestions for Authors
The manuscript titled "Identification and Characterization of Viral and Bacterial Pathogens in Free-Living Bats of Kopaonik National Park, Serbia" presents a valuable contribution to the field of zoonotic disease research, particularly in the context of bat-associated pathogens. However, there are several areas where the manuscript could be improved to enhance its clarity, scientific rigor, and overall impact.
The introduction part should consider adding a brief mention of the ecological and environmental factors unique to Kopaonik National Park that may influence pathogen prevalence in bat populations.
The sampling period is mentioned as "summer and autumn of 2024," which seems to be a typographical error given the current year. Please clarify the correct sampling period.
The authors should provide more details on the criteria used for selecting the specific bat species and sampling sites within Kopaonik National Park.
It would be beneficial to include information on the sample size calculation and whether it was sufficient to detect low-prevalence pathogens.
The detection of alphacoronaviruses in four samples is noteworthy, but the authors should discuss the potential implications of these findings for public health, especially in light of the high similarity to strains found in Russia. Comparing the findings with similar studies from other regions, particularly in Europe, to highlight any unique or contrasting patterns
Discussing the limitations of the study, such as the relatively small sample size and the potential for false negatives due to the sensitivity of the molecular methods used.
Providing recommendations for future research directions, such as longitudinal studies to monitor pathogen dynamics over time or studies to investigate the role of environmental factors in pathogen transmission.
Author Response
Dear reviewer,
We appreciate the time and effort that you dedicated to providing feedback on our manuscript and are grateful for the insightful comments on and valuable improvements to our paper.
We accepted all your suggestions and suggestions from other reviewers, we have proofread the entire paper by professional, and we think that the manuscript is now improved.
- The introduction part should consider adding a brief mention of the ecological and environmental factors unique to Kopaonik National Park that may influence pathogen prevalence in bat populations.
We agree with your comment. One paragraph is added in Introduction part.
2. The sampling period is mentioned as "summer and autumn of 2024," which seems to be a typographical error given the current year. Please clarify the correct sampling period.
We agree with your comment. There were three sampling visits, and the dates are inserted in the text.
3. The authors should provide more details on the criteria used for selecting the specific bat species and sampling sites within Kopaonik National Park.
Since there were no previous studies about abundence of bat species, the aim was to capture as many bats as possible.
4. It would be beneficial to include information on the sample size calculation and whether it was sufficient to detect low-prevalence pathogens.
As in previous suggestion, since we did have any information about number of species and number of bat colonies, there was idea to capture as many bats as possible.
5. The detection of alphacoronaviruses in four samples is noteworthy, but the authors should discuss the potential implications of these findings for public health, especially in light of the high similarity to strains found in Russia. Comparing the findings with similar studies from other regions, particularly in Europe, to highlight any unique or contrasting patterns
We agree with your comment and we have answered this question in Discussion section.
6. Discussing the limitations of the study, such as the relatively small sample size and the potential for false negatives due to the sensitivity of the molecular methods used.
We agree with your comment. It is inserted now in the Discussion section.
7. Providing recommendations for future research directions, such as longitudinal studies to monitor pathogen dynamics over time or studies to investigate the role of environmental factors in pathogen transmission.
The manuscript has been proofread by professional and we beleave that now it is much better.
Best regards,
Authors
Reviewer 4 Report
Comments and Suggestions for Authors
Dear Authors,
Thank you for submitting your manuscript for review. While your study on the detection of viral and bacterial pathogens in free-living bats of Kopaonik National Park addresses an interesting and relevant topic, I have several concerns regarding its scientific rigor and methodology.
One primary issue is that the detection of nucleic acids from oral and anal swab samples using PCR/qPCR techniques does not provide sufficient evidence to confirm the presence, detection, or carriage of these pathogens in bats. The detected nucleic acids could merely be residual genetic material rather than indicative of viable pathogens. Without direct pathogen isolation, it is not possible to conclude that these agents are actively present or being carried by bats.
Additionally, the methodology section lacks sufficient detail. It does not clearly specify the source and isolation method for each pathogen, nor does it provide explicit information on the primers used for bacterial identification or the reliability of the employed techniques. This lack of clarity makes it difficult for readers to assess the validity of the findings without consulting external literature, which is not suitable for a research article.
Another significant concern is the reliance on a single swab sample for the presence/absence analysis of certain pathogens using PCR. This approach does not align with the expected standards for a research article. Furthermore, both the methods and results sections are extremely brief, further limiting the scientific impact of the study.
Author Response
Dear reviewer,
We appreciate the time and effort that you dedicated to providing feedback on our manuscript and are grateful for the insightful comments on and valuable improvements to our paper.
We accepted all your suggestions and suggestions from other reviewers, we have proofread the entire paper by professional, and we think that the manuscript is now improved.
- One primary issue is that the detection of nucleic acids from oral and anal swab samples using PCR/qPCR techniques does not provide sufficient evidence to confirm the presence, detection, or carriage of these pathogens in bats. The detected nucleic acids could merely be residual genetic material rather than indicative of viable pathogens. Without direct pathogen isolation, it is not possible to conclude that these agents are actively present or being carried by bats.
We agree with your comment. However, oral and anal swabs were taken from live bats. Bats are a protected species in the Republic of Serbia and their killing is not permitted. During the live capture of bats, we did not find any dead bats to attempt to detect pathogens from their organs. After swabbing, the bats were released back into the wild. Molecular techniques for pathogen detection were used, in addition to being faster and more economical, do not require biological safety level 3 (BSL-3) or BSL-4 laboratories, which are not available in the Republic of Serbia. Isolation of lyssaviruses and filoviruses would require such a facilities.
- Additionally, the methodology section lacks sufficient detail. It does not clearly specify the source and isolation method for each pathogen, nor does it provide explicit information on the primers used for bacterial identification or the reliability of the employed techniques. This lack of clarity makes it difficult for readers to assess the validity of the findings without consulting external literature, which is not suitable for a research article.
We agree with your comment. A table with primers and probe sused in this study is added to the text.
- Another significant concern is the reliance on a single swab sample for the presence/absence analysis of certain pathogens using PCR. This approach does not align with the expected standards for a research article. Furthermore, both the methods and results sections are extremely brief, further limiting the scientific impact of the study.
We have expanded the parts you requested. The manuscript has been proofread by professional and we beleave that now it is much better.
Best regards,
Authors
Reviewer 5 Report
Comments and Suggestions for Authors
Line 26: investigated
Line 39 : instead “preparedness”, use “prevention”
Line 40: emerging zoonotic diseases
Line 41 - instead “bats”, use “Chiroptera”, because “bats” is already in the title
Line 45 - ...and emergence of zoonosis - diseases that...
Line 69: ...MERS-CoV. A decade....
Line 75: ...populations [8].
Line 94: delete ")"
Line 100: ..."this" country
we don´t use possessive pronouns in scientific texts.
Lines 120-121: It should be interesting to write were this permission can be consulted (link, home page ecc...)
Line 121: Protection
Lines 131 to 133: You have to cite the bibliography of this method. If you have not this bibliography, describe this method here, or describe it.
Line 181: In scientific language, don´t use possessive pronouns
Instead "Our" use "This"
Line 198: nine
Line 202: “These” results..
WE DON´T USE POSSESSIVE PRONOUNS IN SCIENTIFIC TEXTS.
Lines 212-214: what is the bibliographic citation?
Lines 217 to 227: there is a lot of information here. You should cite the bibliography.
Lines 248-249: we don´t use possessive pronouns in scientific texts.
Line 299: Have you observed in the captured bats, ticks, or another kind of ectoparasites that can act as vectors??
This information should be quite interesting
Line 304: deelete "d,"
Line 309: tabulation

Author Response
Dear reviewer,
We appreciate the time and effort that you dedicated to providing feedback on our manuscript and are grateful for the insightful comments on and valuable improvements to our paper.
We accepted all your suggestions and suggestions from other reviewers, we have proofread the entire paper by professional, and we think that the manuscript is now improved.
Please note that line numbering has changed now.
Comments and Suggestions for Authors
Line 26: investigated
Line 39 : instead “preparedness”, use “prevention”
Line 40: emerging zoonotic diseases
Line 41 - instead “bats”, use “Chiroptera”, because “bats” is already in the title
Line 45 - ...and emergence of zoonosis - diseases that...
Line 69: ...MERS-CoV. A decade....
Line 75: ...populations [8].
Line 94: delete ")"
Line 100: ..."this" country
we don´t use possessive pronouns in scientific texts.
We agree and all above comments are corrected.
Lines 120-121: It should be interesting to write were this permission can be consulted (link, home page ecc...)
Unfortunately, there is no possibility for that.
Line 121: Protection
We agree and we have corrected all above.
Lines 131 to 133: You have to cite the bibliography of this method. If you have not this bibliography, describe this method here, or describe it.
We agree, the metod is added to the referenced and it is briefly described in materials and methods.
Line 181: In scientific language, don´t use possessive pronouns
Instead "Our" use "This"
Line 198: nine
Line 202: “These” results..
WE DON´T USE POSSESSIVE PRONOUNS IN SCIENTIFIC TEXTS.
We agree and we have corrected it.
Lines 212-214: what is the bibliographic citation?
The citation is inserted.
Lines 217 to 227: there is a lot of information here. You should cite the bibliography.
All the information are form two references that are stated.
Lines 248-249: we don´t use possessive pronouns in scientific texts.
Corrected.
Line 299: Have you observed in the captured bats, ticks, or another kind of ectoparasites that can act as vectors??
This information should be quite interesting
Interestingly, we did not noted ectoparasites in the bats.
Line 304: deelete "d,"
Line 309: tabulation
Corrected.
Please note that the manuscript has been proofread by professional and we beleave that now it is much better.
Best regards,
Authors
Round 2
Reviewer 2 Report
Comments and Suggestions for Authors
The quality of the current manuscript has been improved greatly. I now agree for further process. Congratulations!
Author Response
Thank you for your careful evaluation of our manuscript and for your valuable suggestions. It have improved it significantly.
Best regards,
Authors
Reviewer 4 Report
Comments and Suggestions for Authors
Dear Authors,
While I acknowledge the revisions made to the manuscript, I remain concerned that the study does not present high-quality data regarding bacterial and viral isolation and identification. The primary reason for this is the methodological limitation stemming from the use of oral and anal swabs for targeted microorganism detection. If the aim is to investigate the microbial carriage of these bat species, a more appropriate approach would be shotgun metagenomic analysis of the collected swabs. Such an analysis would comprehensively reveal the entire microbial load, thereby identifying all microorganisms carried by these species.
In its current form, I find the scope of the study to be narrow, lacking in scientific novelty, and not supported by sufficiently robust data.
Author Response
Comments:
While I acknowledge the revisions made to the manuscript, I remain concerned that the study does not present high-quality data regarding bacterial and viral isolation and identification. The primary reason for this is the methodological limitation stemming from the use of oral and anal swabs for targeted microorganism detection. If the aim is to investigate the microbial carriage of these bat species, a more appropriate approach would be shotgun metagenomic analysis of the collected swabs. Such an analysis would comprehensively reveal the entire microbial load, thereby identifying all microorganisms carried by these species.
In its current form, I find the scope of the study to be narrow, lacking in scientific novelty, and not supported by sufficiently robust data.
Response:
Dear Reviewer,
Thank you for your careful evaluation of our manuscript and for your valuable suggestions. We acknowledge the methodological limitations you highlighted and have made efforts to address them through additional analyses and clarifications in the revised version.
As you noted, the sample size is relatively small, which is primarily due to the unique ecological conditions of Kopaonik National Park. At altitudes above 1,000 meters, bat population densities are significantly lower compared to lowland areas. This challenge was further compounded by the lack of prior systematic research on bat distribution and potential roosting sites within the park. Given these constraints, we believe that the identification of 12 distinct bat species represents a meaningful contribution to the understanding of Chiroptera diversity, not only in Serbia but also in the wider region.
We particularly emphasize the detection of alphacoronavirus in Myotis mystacinus—a finding that, to our knowledge, has not been previously documented in the literature. This discovery could have important implications for further research on the zoonotic potential of these viruses.
Regarding molecular analyses, we fully recognize their limitations, which were largely dictated by the project’s budgetary framework. Nevertheless, we believe this pilot study provides a solid foundation for future research, including more comprehensive metagenomic studies with larger sample sizes and advanced bioinformatic processing.
We hope that our revisions and responses have adequately addressed your concerns, and that the manuscript will now be deemed suitable for publication.
Sincerely,
Authors